# Hybrid Sausages: Modelling the Effect of Partial Meat Replacement with Broccoli, Upcycled Brewer’s Spent Grain and Insect Flours

**DOI:** 10.3390/foods11213396

**Published:** 2022-10-27

**Authors:** Clara Talens, Raquel Llorente, Laura Simó-Boyle, Isabel Odriozola-Serrano, Itziar Tueros, Mónica Ibargüen

**Affiliations:** 1AZTI, Food Research, Basque Research and Technology Alliance (BRTA), Parque Tecnológico de Bizkaia, Astondo Bidea, Edificio 609, 48160 Derio, Bizkaia, Spain; 2Department of Food Technology, University of Lleida—Agrotecnio Center, Rovira Roure 191, 25198 Lleida, Lleida, Spain

**Keywords:** hybrid meat, sustainable nutrition, brewer’s spent grain, insects, upcycling, statistical optimisation

## Abstract

The social, environmental and health concerns associated with the massive consumption of meat products has resulted in calls for a reduction in meat consumption. A simplex lattice design was used for studying the effect of combining broccoli, upcycled brewer’s spent grain (BSG) and insect flours from *Tenebrio molitor* (IF) as alternative sources of protein and micronutrients, in hybrid sausages formulation. The techno-functional properties of the ingredients and the nutritional and textural properties of nine hybrid sausages were analysed. The effect of adding these ingredients (constituting 35% of a turkey-based sausage) on protein, fat, fibre, iron and zinc content, and textural properties (Texture Profile Analysis (TPA) and Warner–Bratzler parameters) were modelled employing linear regression (0.72 < R^2^ < 1). The “desirability” function was used for multi-response optimisation of the samples for the highest protein content, optimum chewiness and *a** value (closeness to red). The analysis of sensory data for the three optimised samples showed no significant differences in juiciness and odour between the hybrid meat sausage with 22% broccoli, 3% BSG, and 10% IF and the commercial Bratwurst sausage elaborated exclusively with animal protein. Colour, appearance, chewiness and pastiness were rated higher than for the reference. The instrumental chewiness highly correlated with sensorial chewiness (R^2^ = 0.98). Thus, a strategy introducing less refined and more sustainable sources of protein and micronutrients was successfully employed to model and statistically optimise a meat product formulation with reduced animal protein content.

## 1. Introduction

The social, environmental and health concerns associated with the massive consumption of meat products has resulted in calls for a reduction in meat consumption [1]. Diets including a large proportion of meat have also been linked to an increasing prevalence of cardiovascular, coronary and cerebrovascular diseases, stroke, diabetes type 2 and colorectal cancer [2].

Despite these detrimental effects, many consumers are strongly attached to the traditional, meat-rich diet [3].

Meat is a high-protein food providing all the essential amino acids [4]. It is also a good source of minerals such as selenium and zinc [5]. However, although highly nutritious, it is deficient in dietary fibre [6]. Meeting the nutritional needs while reducing the environmental impact is the overarching concern for future food production and agriculture. The available data, supplied by research in many fields, highlight the importance of incorporating new protein sources into our diet. However, to achieve this goal, we will have to overcome certain engrained attitudes deeply rooted in meat-eating cultures [7]. According to de Boer and Aiking [8], two interrelated types of pro-environmental behaviour relevant to protein consumption are “using fewer natural resources” and “doing things in a different way and with reduced environmental impact”. Examples include reducing the size of meat portions, increasing their nutrient density and replacing meat with plant-based protein sources, such as legumes and nuts.

Comminuted meat products (commonly prepared by cutting, shredding, grounding and mincing) might blend the animal- and plant-based ingredients in what is known as hybrid meat products [9]. Grasso and Jaworska [9] have suggested that it might be more important to persuade the majority of the population to reduce meat consumption than to convince a small percentage to give up meat entirely. Thus, a good practical way to encourage a healthy-eating lifestyle is to introduce the concept of hybrid meat products, in which the vegetables partially replace meat [10].

Hybrid products, created with the idea to facilitate the transition from a meat-based diet to increased plant-based food intake, might assuage the desire for the familiar taste and texture of meat while providing a superior nutritional profile.

There is a conceptual difference between hybrid meat products and meat products with plant-based ingredients. The plant-based functional ingredients are used for economic and technological reasons. The idea of hybrid meat products includes positive connotations, including healthiness, reduced environmental impact and the generally positive effects of decreasing meat consumption.

However, few studies have investigated consumer attitudes towards hybrid meat products. A study by Schösler, Boer [11] has compared various hybrid meat products with alternative proteins made using insects, lentils and seaweed. The most popular was the hybrid snack (chosen by 54% of 1083 participants). The authors have concluded that combining the animal and plant-based proteins might be advantageous; such hybrid meat products could become gradually more acceptable to consumers who would not actively seek environmentally friendly proteins. Similarly, the previous studies by the same authors have found that hybrid meat blends appear familiar and thus could be acceptable to some consumers, especially those who are weakly involved.

One of the factors affecting consumer acceptance of this kind of produce, apart from the food choice motives [12], their attitude towards alternative proteins [13] and food neophobia [14], is the lack of taste compared to conventional meat products. Indeed, it is a commonly held belief among meat eaters that manufacturing healthier versions of meat products or meat analogues compromises their taste [15], lowering sensory expectations [16].

Grasso and Jaworska [9] have shown that lack of taste is one of the most important and frequently reported factors associated with hybrid meat products. Thus, sensory quality is still key in the development of these products. Profeta, Baune [17] have compared burgers made with conventional meat with two types of hybrid meat burgers (with different meat-to-vegetable ratios) and a plant-based burger. They have reported the highest preferences and willingness to pay for the meat burger, followed by the hybrid burger with a higher percentage of meat, the hybrid burger with a lower percentage of meat and, in the last place, the plant-based burger.

Replacing meat with vegetables has been a widely used strategy to improve the nutritional profile of meat products. However, foods with a percentage of substitution below 50% are not considered hybrid products. Broccoli (*Brassica oleracea* var. *italica*) offers a suitable combination of vitamins and minerals, and it can compete with beef and chicken eggs in protein composition and tryptophan and lysine content [18]. Its bioactive phytochemical and nutritional richness, including phenolic and organosulphur compounds, as well as essential vitamins and minerals has called the attention of the scientific community. Furthermore, the broccoli fibre and fibre fetlock regulate the functioning of the gastrointestinal tract. Nazarova, Lazutina [18] has formulated a new type of meat product including broccoli, obtaining a functional, reduced-calorie product. However, there are not many other formulas including broccoli in the hybrid meat products.

Few studies have used upcycled ingredients (vegetable by-products transformed into ingredients for human consumption [19]) or alternative proteins (insect flour, single-cell protein, etc.) in comminuted meat products [20]. Brewer’s spent grain has been recently used as a source of protein in aquaculture [21]. Bjerregaard, Charalampidis [22] has upcycled this ingredient to obtain a food-grade component, and Chin, Chai [23] has also used solid-state fermentation of BSG in food applications.

Kim, Setyabrata [24] have reported that replacing lean pork with insect flours in the formulation of sausages greatly increases the protein content. Although consumers do not favour novel protein sources such as insects, their acceptance increases when insects processed into flour are used as food ingredients [25].

The main aim of this study was to develop a food design strategy to reduce animal protein content in processed meat products. The effect of combining three sustainable sources of protein, fibre and micronutrients on the techno-functional, nutritional, textural and sensory properties of hybrid meat products is modelled and analysed.

## 2. Materials and Methods

### 2.1. Raw Materials

Turkey meat (21.9 g of protein and 2.2 g of fat per 100 g of meat), frozen broccoli (4.4 g of protein, 0.9 g of fat per 100 g, 2.8 g of fibre per 100 g), canned fried onion, refined crystal salt, sweet paprika, black pepper, parsley and olive oil (0.4% acidity) were purchased at a Makro store (Erandio, Spain).

Insect flour was obtained from Insekt Label Biotech (Derio, Spain). The mealworm (*Tenebrio molitor*) composition per 100 g was 52.2 g of protein, 33.0 g of fat and 3.7 g of fibre and 11.1 g of moisture). Brewer’s spent grain (BSG) was obtained using the method described San Martin, Orive [21]. It contained, per 100 g of the product, 22.8 g of protein, 7.7 g of fat, 17.3 g of fibre and 8.0 g of moisture. Natural sausage casings (Apasa, Renteria, Spain) of 24 mm diameter were used.

### 2.2. Physicochemical Analysis of Flours

The particle size distribution of dried BSG and insect flour was measured employing a Static Light Scattering Instrument Master-Sizer 3000 (Malvern Instruments Ltd., Malvern, UK) using bi-distilled water as a dispersion agent (refractive index = 1.33), following a method described by Talens, Lago [26]. The maximum particle diameter of 50% of the sample volume, d_50_ (µm), also known as the median particle size by volume, was obtained. The most common percentiles reported were the d_10_ (µm), d_50_ (µm) and d_90_ (µm). The volume moment mean or De Brouckere mean diameter (D[4.3]) was also shown as it reflected the size of the particles constituting the bulk of the sample volume. It is most sensitive to the presence of large particulates in the size distribution.

The swelling capacity of the flours was determined according to Robertson, de Monredon [27]. For the water solubility index (WSI) of flours, a sample of 2 g was dispersed in 100 mL of distilled water in a water bath at 80 °C for 30 min. The sample was then centrifuged at 1100× *g* for 10 min at room temperature, and the supernatant was collected carefully and dried at 103 ± 2 °C to determine its solid content. The WSI was expressed as the percentage of the original sample present in the soluble fraction (Equation (1)):(1)$$WSI\;\left(\%\right)=\left(Weight\;of\;dissolved\;solid\;in\;supernatant\right)/\left(Weight\;of\;dy\;solids\right)\times 100.\;$$

Water holding capacity (WHC) was measured using Equation (2):(2)$$WHC=\left(\left(Wet\;sample\;weight-dy\;sample\;weight\right)-weight\;of\;water\;retained\;by\;disc\right)/\left(sample\;weight\right)\times 100.\;$$

Oil holding capacity (OHC) was measured using Equation (3).
(3)$$OHC=\left(\left(Wet\;sample\;weight-dy\;sample\;weight\right)-weight\;of\;water\;retained\;by\;disc\right)/\left(sample\;weight\right)\times 100.\;$$

The colour of the samples was measured using a colourimeter (CR-400 Chroma Meter Konica Minolta Inc., Tokyo, Japan) in the CIE *L*a*b** system. The colorimetric parameters *L** (lightness), *a** (redness/greenness) and *b** (yellowness/blueness) were determined.

### 2.3. Hybrid Sausages

#### 2.3.1. Experimental Design and Hybrid Sausage Preparation

A three-factor simplex lattice design with three levels was used to formulate all the possible mixtures (9 assays) containing broccoli (BR), dried BSG and insect flour (IF), adding up to 35% of the total weight of the ingredients (Figure 1).

Different sausage formulations were labelled with F1 to F9 (Figure 2). All 9 combinations contained 400 g of frozen turkey meat, 97 g of canned fried onion, 91.5 g of cold water, 36.6 g of olive oil, 6.4 g of salt, 5 g of carrageenan, 5 g of carboxymethylcellulose (CMC), 1.8 g of parsley, 0.9 g of sweet paprika and 0.9 g of black pepper. The remaining 35% was composed of a 350-g mixture of broccoli, insect flour and/or BSG in different proportions. All ingredients were weighed using high-precision (±0.0001 g) scales AB304-S (Mettler Toledo, Greifensee, Switzerland) and then mixed in a Thermomix food processor (Thermomix TM5; Vorwerk, Wuppertal, Germany) for 20 s at speed 6. The turkey and broccoli were first cut and blended separately, then mixed with the onion and blended again. This was followed by introducing the spices, salt and variable ingredients to the mixture and blending at the same speed (Stephan Mixer UMC 5 Electronic, Stephan Machinery, Hameln, Germany). Then, the carrageenan and CMC were added and mixed in at number 3 speed for 20 s. The ice water and olive oil were mixed in at speed number 6 for 1 min. The mixture was placed in a plastic vacuum bag, the air was removed, and the mixture was introduced into the manual sausage stuffer (TP-10, Mainca, Granollers, Spain). The sausages were first treated at 85 °C for 10 min to stimulate the gelling process (involving carrageenan and CMC), then cooled at room temperature for 15 min, vacuum-packaged in sterilisation bags and labelled. They were pasteurised at 90 °C for 30 min in an combi oven combining heat and vapour treatment (Fagor HEP, Arrasate, Spain).

The temperatures were recorded using a TrackSense Pro Mini Wireless Data Logger Serial No. 84633 (Ellab, Hillerød, Denmark) programmed by the ELLAB software ValSuite Basic 3.1.3.10v (Ellab, Hillerød, Denmark). Data acquisition was performed in intervals of 1 min. In each batch, 1 sample was punctured with the data logger before sealing the bag. The temperature at the core of the product was maintained at 85–88 °C during 17 min, achieving a Fo of 1200. After processing the sausages to the required Fo value, they were cooled rapidly to 4 °C by placing them in the refrigeration chamber. The sausages were stored in a refrigerator at 4 °C and later (after at least 24 h and before at most 72 h) frozen at −20 °C for further characterisation.

#### 2.3.2. Physicochemical Characterisation of Hybrid Sausages

The nutritional content of the hybrid sausages was analysed following the ISO standards for moisture (ISO 1442:1997), ash (ISO 936:1998), protein content (ISO 937:1978), crude fat (ISO 1443:1973), and iron and zinc (DIN EN 16943:2017-07). The total dietary fibre (TDF), soluble dietary fibre (SDF) and insoluble dietary fibre (IDF) were determined using the AOAC enzymatic gravimetric method, 991.43. After proximate analysis, energy values were calculated using the Atwater general factors (4 kcal/g for protein and carbohydrates, 9 kcal/g for fat and 2 kcal/g for fibre). Samples were analysed in triplicate.

The colour of the samples was measured using the colourimeter in the CIE *L*a*b** system. The colorimetric parameters *L** (lightness), *a** (redness/greenness) and *b** (yellowness/blueness) were determined.

#### 2.3.3. Textural Properties

The texture properties of cooked sausages were evaluated employing the Texture Profile Analyses (TPA) and Warner–Bratzler (WB) test, using an HD-Plus texturometer (Stable Micro System, Godalming, UK) attached to a 50-N load cell.

The TPA was performed as described by Bourne and MC [28], using a 75-mm compression platen probe (P/75). Samples (∅ = 2.4 cm, height = 2 cm, at 4 °C) were axially compressed twice to 25% of their original height, avoiding fracture. Force–time deformation curves were obtained at a crosshead speed of 1 mm/s and a time interval of 5 s between the two compression cycles. The following attributes were calculated: hardness (N), adhesiveness (N.mm), cohesiveness (dimensionless), springiness (dimensionless), chewiness (N) and resilience (dimensionless). Three samples per batch were examined, each batch in duplicate. The results of these 6 measurements were then averaged for each formula.

The WB test was performed using a V-shaped blade. Sausages (12 mm × 2.4 cm, 4 °C) were cut transversally at a constant crosshead speed of 2 mm/s. The work needed to move the blade through the sample (work of shear) was recorded in N.mm. This test was performed in triplicate per batch, and each batch was tested in duplicate. The results of these 6 measurements were then averaged for each formula.

#### 2.3.4. Sensory Analysis

A quantitative descriptive analysis (QDA) was performed to assess the sensory characteristics of the hybrid sausages. A preliminary recall training was preceded by four sessions at which the members of a trained panel of 8 assessors (7 women and 1 man) were re-trained to identify and evaluate sensory characteristics of a commercial sausage (100% meat) of the same dimensions and appearance.

They were also trained in the use of a 0–9 intensity scale. In these sessions, the participants generated the descriptors and their definitions. They were asked to name the attributes that they considered important for the descriptive evaluation of cooked sausages (“Bradwurst” type) and formulate their definitions in open discussion. For the training session, a commercial Bratwurst sausage from a Spanish supermarket was used.

Finally, a QDA scale was agreed upon, based on the commercial control, to place a sample as close as possible to the values obtained for the QDA in the following sessions. The trained assessors generated six attributes described below.

Appearance, including the presence of spots and solid chunks. The absence of spots or pieces was rated as 0.

Colour: White was assessed as 1, mushroom colour as 6, and dark brown as 9.

Odour: Characteristic smell of fresh butcher’s sausage was rated as 1, and of Bratwurst sausage (cooked sausages), as 9.

The chewiness was expressed as the number of chews needed before swallowing a standard bite of the commercial reference sausage (Bratwurst). The number was different for each assessor.

Juiciness: Chickpea paste was used as the reference for the lowest juiciness (1), whereas the fresh butcher’s sausage was considered the juiciest (9).

Pastiness: Broth was the reference for the lowest pastiness (1), and peanut butter for the highest level of pastiness (9).

After training, a statistical analysis was conducted to identify the discrimination capacity, repeatability and reproducibility of each panellist, as well as the robustness of the panel. Each panellist evaluated 5 samples presented in random order. This robust, trained panel conducted all sensory evaluations.

Each assessor was given a plate of 5 samples, labelled with 3-digit random numbers, for evaluation. The samples were cooked for 3 min on a simple electric hob, using a frying pan. The attributes were assigned values on a 0–9 intensity scale. All samples were assessed in random order in three replicates. The panellists graded each sample on the basis of the intensities of generated attributes. The results were analysed to see if the products differed significantly and which attributes made them different.

### 2.4. Statistical Analysis and Modelling of Experimental Data

The experimental design, modelling and optimisation were conducted employing the R-project software v 4.1.2 (R Foundation for Statistical Computing, Vienna, Austria). The packages used were “readxl”, “tidyverse”, “mixexp”, “rsm”, “ggtern”, “Ternary”, “ggplot2”, “agricolae”, “desirability” and “fsmb”.

The polynomial equation used was:Y = β_1_ X_1_ + β_2_ X_2_ + β_3_ X_3_ + β_12_ X_1_X_2_ + β_13_ X_1_ X_3_ + β_23_ X_2_ X_3_ + β_123_ X_1_ X_2_ X_3_(4)
where Y is the estimated response and β_1_, β_2_, β_3_, β_12_, β_13_, β_23_ and β_123_ are constant coefficients for each linear and nonlinear (interaction) term produced for the prediction models of processing components. The computational work was performed using the R-project (v 4.1.2) “ternary” package. The model for each response was obtained based on the fitting quality, the coefficient of determination (R^2^) and the significant level of regression (*p* < 0.05). A one-way analysis of variance (ANOVA) and a Tukey HDS test were used to determine pairwise differences between groups (*p* < 0.05). The “desirability” package was used for the optimisation of experimental design to obtain the maximum protein content (from a nutritional perspective), maximum chewiness (from a textural perspective) and maximum *a** (greenness to redness on the *L*a*b** scale, considering closeness to the colour of 100%-meat sausage).

## 3. Results

### 3.1. Physicochemical Characterisation of Protein Ingredients

The functional properties, colour parameters and particle size of BSG and IF are summarised in Table 1.

The moisture content of BSG and IF was 7.0 and 6.0%, respectively, with no significant differences (*p* > 0.05) between the two ingredients.

Swelling capacity was significantly lower (*p* < 0.05) for the IF (3.40 mL/g) than for BSG (4.70 mL/mg); the latter showed the highest value. There were no significant differences (*p* > 0.05) between the water solubility indices, water holding capacities and oil holding capacities.

The differences between the swelling properties of the BSG and IF might be due to particle size differences. Indeed, the two powder ingredients studied mainly contained particles of average diameter (d_50_) from 372.89 µm for BSG to 481.56 µm for IF. It is important to note that for IF, d_90_ (854.11 µm) was significantly larger (*p* < 0.05) than for BSG (766.44 µm), indicating that this product contained coarse particles larger than 700 µm, which can influence swelling properties during hydration.

The coarse particles (D[4.3]) were significantly larger for the IF (468.11 µm) than for the BSG (407.67 µm). These differences might have resulted from the disparate milling procedures; the size of the mesh (unknown to the authors) would affect the size of the flour particles and, therefore, their technical properties.

The colours of the ingredients were significantly different (*p* < 0.05) (Table 1). The BSG had the highest *L** value (57.22), and the IF had the darkest *L** value (27.74). Moreover, *a** and *b** also significantly differed (*p* < 0.05) between these two ingredients; the IF showed higher *a** values (8.92) than the BSG (5.88). Positive *a** values indicate closeness to red, whilst negative values indicate colours closer to green. Large positive *b** values are associated with closeness to yellow. The IF ingredient had a higher *b** value (25.87) than the BSG (24.38).

### 3.2. Nutritional Content and Colours of the Hybrid Sausages

The mean values for the nutritional properties and colourimetric parameters of the different cooked hybrid meat sausages can be found in Table 2.

The protein content varied from 13.4 to 18.1%. The F1 samples, with the largest proportion of broccoli, had the lowest protein content. The samples in which broccoli was substituted by IF and/or BSG powders contained more than 14.1% of protein concentrations. This increase in protein levels reflects the large proportion of protein in the IF (52.2 g/100 g) and BSG powders (22.8 g/100 g) in comparison with the broccoli (4.4 g/100 g). According to the EU regulations, the source-of-protein claim can be used when proteins provide at least 12% of the energy. It should be noted that all hybrid meat samples were good sources of protein (more than 45% of the total energy).

The fat content was more than 5.6 g/100 g in all the samples. Increasing the level of substitution of broccoli with dry ingredients also increased the fat content. Moreover, F3, F4 and F8, containing the largest proportion of IF (10%), had the highest concentration of fat (8.4–8.7%). This was due to very high levels of this macronutrient in the IF (33%).

The fibre content was above 3 g/100 g for F2, F4, F5, F6 and F7. Thus, using a large proportion of broccoli (30–32%) in combination with the BSG or IF or adding 5% BSG and IF and 25% of broccoli makes the hybrid meat sausages a good source of fibre. The EU regulations state that a source-of-fibre claim may be made where the product contains at least 3 g of fibre per 100 g.

Adding the IF or BSG powders to the hybrid meat samples decreased their moisture content and increased the micronutrients (iron and zinc) concentration. However, it should be pointed out that at the same level of broccoli replacement (10%), the content of iron was reduced in samples elaborated with 10% of IF (F3) compared to those containing 5% IF and 5% BSG (F7).

F4, with the smallest addition of crude broccoli (20%) and the largest amounts of BSG (5%) and IF (10%), had the lowest moisture content and the highest concentration of nutrients. The replacement of 15% of broccoli with insect flour led to an increase in solid compounds because the IF and BSG preparations have 90–92% dry matter content (10% for broccoli).

Regarding colour parameters, the samples had similar values of yellowness (*b**), irrespective of the ingredients used in the formulation. Even though the *b** value was significantly different for the two powder ingredients (Table 1), this was not perceived when they represented a small proportion (10% or less) of the product composition.

The F1 samples, with 35% of crude broccoli, had a significantly lower *a** (redness) value (−1.6) than the rest of the samples. The F1, F5 and F6 samples, elaborated with 30% or more of crude broccoli in combination with ≤3% of BSG powder, showed negative *a** values, indicating large amounts of green pigments related to the chlorophyll. In contrast, the F2 sample had positive *a** values (1.4); thus, combining the high content of broccoli (30%) with the BSG at 5% results in sausages with a less green appearance. The reddest samples were those in which 15% of the broccoli was replaced by BSG and IF (F4). This agrees with the *a** values shown for BSG (5.88) and IF (8.92) in Table 1. The larger the proportion of these ingredients in the formula, the higher the redness values (*a**).

The F1 and F6 samples, elaborated with a high concentration of crude broccoli (≥32%), had the highest lightness values (*L**). However, non-significant differences between *L** values were found for the hybrid meat samples where 5–13% of broccoli was replaced by BSG or IF powders. The F4 samples, with substitutions using IF (10%) and BSG (5%), showed the lowest *L** values. The data in Table 2 demonstrate that the IF powder reduces the lightness values (27.74) more than the BSG (57.22). Thus, the IF darkens the colour of the final product when blended with other ingredients.

### 3.3. Textural Properties of the Hybrid Meat Sausages

The mean values of the textural properties of the different cooked hybrid meat sausages can be found in Table 3. All the formulations incorporating the IF or BSG showed a significant increase in hardness and chewiness (Figure 3).

The F1 samples, with 35% of crude broccoli, had a significantly softer texture than the others. In contrast, F4 and F8 samples, with 25% and 22% of broccoli, 5% and 3% of BSG, respectively, plus 10% of IF both, were significantly harder and chewier than others. In this case, raising the amount of dry ingredients substituting the broccoli increased the hardness and chewiness.

No significant differences were found between the values obtained for adhesiveness, springiness or cohesiveness (*p* > 0.05) of different samples.

Resilience is the capability to recover from deformation in terms of speed and force [29]. In simple terms, it measures the elastic recovery of the sample when a compression load is removed. Samples F3 and F5, where only broccoli and IF were used, showed higher resilience than the remaining samples. Thus, adding IF improved the elastic properties of the hybrid meat sausages. The increased protein content in these sausages could explain this relatively high elastic recovery.

However, the resilience values obtained for mixtures of IF and BSG (F4, F7, F8 and F9) were significantly lower than for other samples.

It is worth noting that springiness and resilience are the parameters that reflect the elastic properties of samples. However, in this study, significant differences were found between the data for resilience (*p* < 0.05) and not for springiness.

Shearing analysis revealed that the F4, F8 and F7 samples required significantly (*p* < 0.05) more work to shear than the rest of the samples (Figure 4). The sausages F4, F7 and F8 contained 15, 10 and 13% of dry ingredients, respectively, and they were harder and chewier than the others. Their relatively large work-of-shear values could also be related to an increased proportion of solid compounds. Augmenting the firmness of gels can raise the resistance to compression and shearing, explaining the results obtained in these tests.

Table 4 shows the linear regression results for the different responses. The effect of adding the three different ingredients (or their combinations) is reflected by the F-value and the corresponding *p*-value, explained in the text as (F-value, *p*-value) for each effect. Ternary diagrams are shown for each response in Figure 5. Protein content could be predicted with R^2^ of 1.0 (388.95, *p* < 0.0001). The IF showed the strongest effect in the model (51.71, *p* < 0.001), followed by the combination BR*BSG (26.36, *p* = 0.004), and BR*IF (1.66, *p* = 0.25). The insect flour contained more protein (52.2%) than the BSG (22.8%) or broccoli (4.4%), explaining the stronger effect of this ingredient on the protein content of the hybrid meat sausages.

The R^2^ for fat content was 0.99 (321.55, *p* < 0.0001). The fat content was also highly correlated with the BR and BSG (321.5, *p* < 0.01). The broccoli had the strongest effect (632.76, *p* < 0.0001), followed by the BSG (71.18, *p* = 0.0002).

Fibre content also had an R^2^ of 1 (38,656.32, *p* < 0.0001). Adding the BSG had the strongest effect (60,584.75, < 0.0001) followed by the broccoli (23.69, *p* = 0.003). The fibre content of the BSG was 17.3%, whereas IF supplied 3.7% and broccoli 2.8%.

Iron content (Fe) had a coefficient of regression of 0.74 (4.68, *p* = 0.06), with significant interactions with broccoli and BSG (7.68, *p* = 0.04) and BR*IF (4.13, *p* = 0.1). Zinc content (Zn) had a coefficient of regression of 0.72 (7.84, *p* = 0.02). Broccoli showed a very significant effect (15.01, *p* = 0.008), whereas the BSG effect was much weaker (1.03, *p* = 0.35).

The hardness and adhesiveness regression coefficients were below 0.7, with a *p*-value ≥ 0.05. The values of R^2^ for cohesiveness, work of shearing and shearing force were ≤0.9, and for the springiness, chewiness and resilience, the R^2^ was ≥0.9. To be exact, the R^2^ was 0.935 (23.79, *p* = 0.002) for springiness, 0.96 (40.44, *p* = 0.001) for chewiness and 0.952 (33.07, *p* = 0.001) for resilience.

It is worth mentioning that adding the IF had a significant effect (*p* < 0.05) on cohesiveness and a very significant effect (*p* < 0.01) on chewiness and resilience. The BSG also had a significant effect (*p* < 0.05) on hardness, cohesiveness and work of shearing and a very significant effect (*p* < 0.01) on the shearing force. The broccoli did not significantly affect any of the texture attributes measured.

In the regression analysis of the colour parameters, we observed a strong correlation of the hybrid meat composition with *L**, *a** and *b**. The coefficient of regression for *L** was 0.933 (23.199, *p* = 0.002), with a significant effect of IF (11.284, *p* = 0.02). It was 0.991 (187.545, *p* < 0.0001) for *a**, with a highly significant effect of broccoli (246.62, *p* < 0.0001), IF (134.344, *p* < 0.0001) and BSG*IF (25.77, *p* = 0.004). Lastly, the *b** had the R^2^ value of 0.912 (17.56, *p* = 0.004), and was significantly affected by BR*IF (22.92, *p* = 0.005) and BR*BSG (20.66, *p* = 0.006).

Finally, the “desirability” function was used to select the samples to be assessed by the trained panel. This mathematical function allows optimisation of the models for each response, selecting the maximum, minimum or target desired values. In this study, the desirability function was applied to choose the samples with the maximum protein content, chewiness and a* value. These responses were selected bearing in mind the significance of the regression coefficients obtained and the number of samples that a trained panel can assess simultaneously (usually no more than four samples per session). For this reason, considering that one control was used as commercial reference (100% meat), and another control was F1 (35% broccoli, 0% BSG and 0% IF), the other two samples with the highest desirability were chosen for the sensory analyses. The desirability values were obtained for F1 (0), F2 (0.116), F3 (0.619), F4 (0.998), F5 (0.299), F6 (0.205), F7 (0.658) and F8 (0.966). Samples F4 and F8 were chosen following the above criteria. However, the preliminary tasting sessions showed that the F4 and F7 had an unpleasant taste due to the high content of BSG (5%). Therefore, these samples were excluded from the sensory analyses; the trained panel assessed only samples F3 (desirability of 0.619) and F8 (0.996).

### 3.4. Sensory Results

Table 5 shows the results of the analyses of the sensory data supplied by the trained panel. All samples were obtained higher scores than the commercial Bratwurst sausage used as control (100% meat). REF sample had a score of 5.62 for appearance, whereas F8, F3 and F1 achieved 8.13, 7.73 and 6.73, respectively.

Sample F1 (35% broccoli) obtained the lowest chewiness score. The control sample (REF) and F3 (25% broccoli and 10% IF) did not differ significantly (*p* > 0.05). Sample F8 (22% broccoli, 3% BSG and 10% IF) reached the highest chewiness value (6.87), probably due to the largest content of powdered ingredients (13%) compared to F3 (10%).

Sample F8 (7.8) and F3 (7.20) showed the highest colour intensity (7.8 and 7.20, respectively); these samples were significantly different (*p* < 0.05) from the rest. This effect can be related to the darker colour provided by the insect flour; both samples contained 10% of IF. The control sample (REF) and F1 (35% broccoli only) were not considered significantly different (*p* > 0.05) by the panel. The absence of IF (with its dark colouring) in these two samples explains this result.

Sample F1 (35% broccoli) was considered significantly juicier than F3 (25% broccoli, 10% IF) and F8 (22% broccoli, 3% BSG and 10% IF). The scores for REF and F8 differed slightly (4.38 and 4.43, respectively). This effect can be explained by the high moisture content of the broccoli improving the perception of juiciness. As the amount of broccoli was reduced, juiciness decreased. A similar effect was observed with increasing content of powdered ingredients.

The smell of the control sample and F8 did not differ significantly (7.00 and 6.60, respectively). This implies that combining the three ingredients in the right proportion (22% broccoli, 3% BSG and 10% IF) could produce an aroma similar to a 100%-meat sausage. The odour significantly decreased in the absence of BSG, i.e., in samples F3 (5.90) and F1 (4.8).

The control sample did not obtain a high pastiness score (3.35); this attribute was more noticeable in the hybrid samples (without significant differences between them): 5.67 for F8, 5.27 for F3 and 5.57 for F8.

Figure 6 depicts the results from the sensory analysis. In general, it can be concluded that sample F8 was the most similar to the control in terms of chewiness, juiciness and odour (see mean values for F8 in Figure 6). Furthermore, this sample obtained high scores for appearance and colour. However, it showed an increase in pastiness, probably due to the water retention capacity of the BSG fibres.

## 4. Discussion

The range of moisture content (6–12%) was typical for the powder ingredients obtained from alternative sources such as BSG [21] and insects [26]. The differences in hydration properties can be due to components other than protein, such as starches and fibres, that can trap water molecules. Ahmed, Thomas [30] have shown that, as the particle size of quinoa flours decreases, their rheological parameter values are also reduced (viscosity, rigidity). These changes are associated with a reduction in starch content of smaller particles (lowered water retention potential). The flours used in this study also contained carbohydrates and other molecules contributing to these properties. Kim, Setyabrata [24] have reported that replacing lean pork with insect flours in sausage formulations greatly increases the protein content. A similar increase in fat content, as a result of using insect flours in the elaboration of frankfurters, has been reported in other studies [31]. Kim, Setyabrata [24] have observed that adding mealworm flour significantly increases the amounts of zinc and iron (among other minerals) in lean pork sausages. Sisik, Kaban [32] have reported that adding broccoli to the sausages causes a decrease in the *a** value compared to the formulations without broccoli. Özvural, Vural [33] and Choi, Kim [31] have concluded that the redness of sausages increases due to the addition of BSG to the formulation. In agreement with these results, Choi, Kim [31] have observed a reduced lightness in frankfurters made with the addition of mealworm flour. A decrease in hardness and chewiness due to the substitution of meat with vegetables and the substitution of chicken with carrot and cabbage has been reported in other studies [32,34]. Kim, Setyabrata [24] have concluded that augmented hardness is an inevitable result of reduced moisture content and increased proportion of solid compounds in sausage formulations using pre-treated insect flour. Pereira, Ramos [35] have reported that the final protein content affects the gelation and emulsification processes and improves the textural properties. This effect could also be explained by the insect proteins forming hydrogen bonds during cooling [36]. According to this study, the mixtures of certain proteins, such as insect proteins, might be unable to aggregate during the gelling structure formation, decreasing the gel strength. Shin and Choi [37] have observed variations in significant differences between the values of textural parameters for sausage samples due to changes in compression rate and crosshead speed during the tests. At a low compression rate of 20% (similar to the rate used in this study), they found no significant differences between the springiness of the samples. However, when the compression ratio increased to 30%, significant differences were observed for all the textural parameters (*p* < 0.05). There was a strong correlation (R^2^ = 0.983) between the chewiness values obtained in the instrumental texture test and the chewiness assessed by the panellists. This means that these tests predict the chewiness of hybrid meat sausages perceived by a human panel. Özvural, Vural [33] and Grigelmo-Miguel and Martín-Belloso [38] have reported that the increased BSG concentration in sausages reduces the juiciness scores. Choi, Kim [31] also have reported decreased juiciness of frankfurters after increasing the insect flour content (up to 10%). The BSG volatile compounds, bestowing a pleasant taste and aroma appreciated by consumers, have been identified by Farcas, Socaci [39] in baked goods. The same aroma intensity was noted for the sample F8, where it was rated as similar to the commercial sample with 100% meat.

## 5. Conclusions

This study demonstrates that the inclusion of alternative protein sources can be effective as a strategy to reduce animal protein content in meat product formulations. Moreover, adding alternative ingredients can create synergistic effects; the hybrid meat products with 50% animal protein have textural and sensory properties similar to the commercial formulations elaborated exclusively with animal protein. The three ingredients studied here, the broccoli, upcycled brewer’s spent grain and insect flour (*Tenebrio molitor*), can be used to achieve the textural, nutritional and sensory properties similar or even superior to those of the meat-only products. A mixture design methodology was used to obtain the new formulations, and the nutritional and textural properties were modelled and predicted with a high level of correlation with the ingredient proportions. The statistical optimisation was based on the desirability function. The formulation containing 22% broccoli, 3% BSG and 10% IF had the juiciness and odour comparable to the commercial sample and was superior in terms of appearance. Future work should include the analysis of the fatty acid profile of hybrid sausages, which can be affected by the integration of these sustainable sources of macro- and micro-nutrients. As the ingredients used here were less refined than those from traditional sources of protein, it is important to consider the effect of their inclusion on other relevant components such as fats or micronutrients.

## Figures and Tables

**Figure 1 foods-11-03396-f001:**
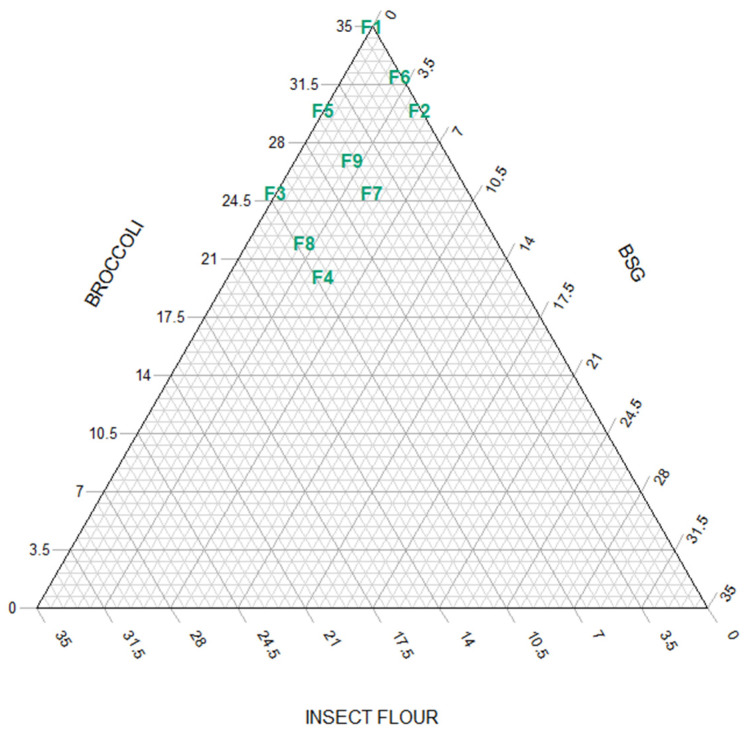
Simplex design plot of ingredient proportions for sausages with replacements adding up to 35%. The samples, labelled from F1 to F9, contain different percentages of broccoli, upcycled brewer’s spent grain and insect flour (BR, BSG, IF): F1 (35, 0, 0), F2 (30, 5, 0), F3 (25, 0, 1), F4 (20, 5, 10), F5 (30, 0, 5), F6 (32, 3, 0), F7 (25, 5, 5), F8 (22, 3, 10) and F9 (27, 3, 5).

**Figure 2 foods-11-03396-f002:**
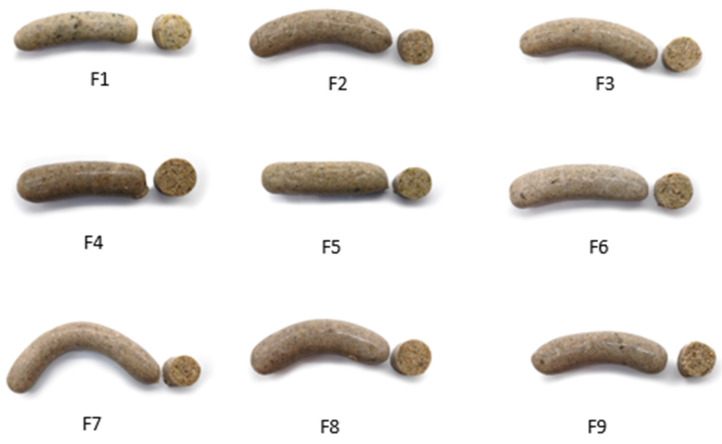
Images of the 9 hybrid sausages used in the experiments. The samples, labelled from F1 to F9, contain different percentages of broccoli, upcycled brewer’s spent grain and insect flour (BR, BSG, IF): F1 (35, 0, 0), F2 (30, 5, 0), F3 (25, 0, 1), F4 (20, 5, 10), F5 (30, 0, 5), F6 (32, 3, 0), F7 (25, 5, 5), F8 (22, 3, 10) and F9 (27, 3, 5).

**Figure 3 foods-11-03396-f003:**
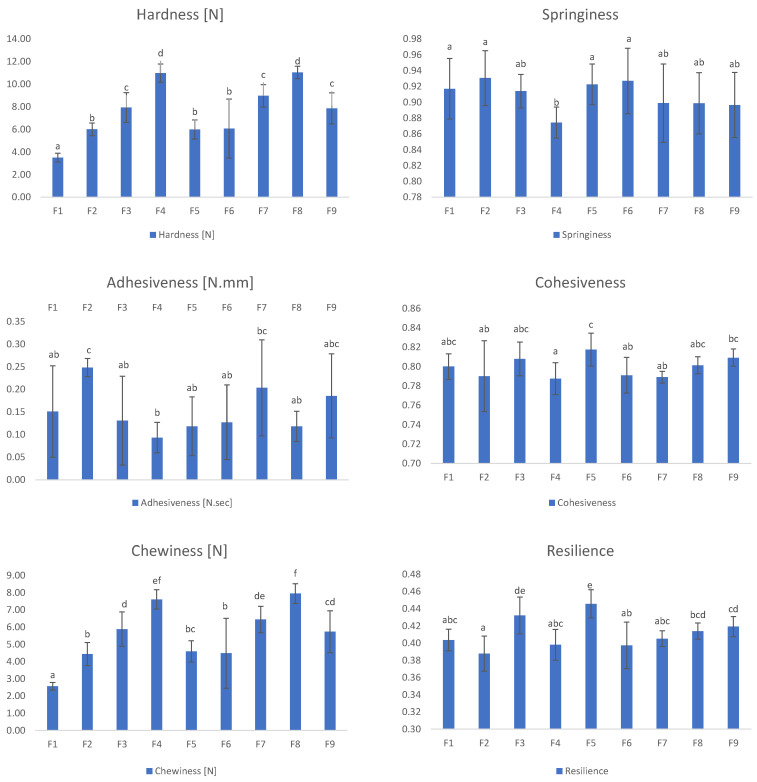
Texture Profile Analyses (TPA) of the 9 hybrid sausages used in the experiments. The samples, labelled from F1 to F9, contain different percentages of broccoli, upcycled brewer’s spent grain and insect flour (BR, BSG, IF): F1 (35, 0, 0), F2 (30, 5, 0), F3 (25, 0, 10), F4 (20, 5, 10), F5 (30, 0, 5), F6 (32, 3, 0), F7 (25, 5, 5), F8 (22, 3, 10) and F9 (27, 3, 5). Different letters in the same bar are significantly different according to Tukey’s test (*p* < 0.05).

**Figure 4 foods-11-03396-f004:**
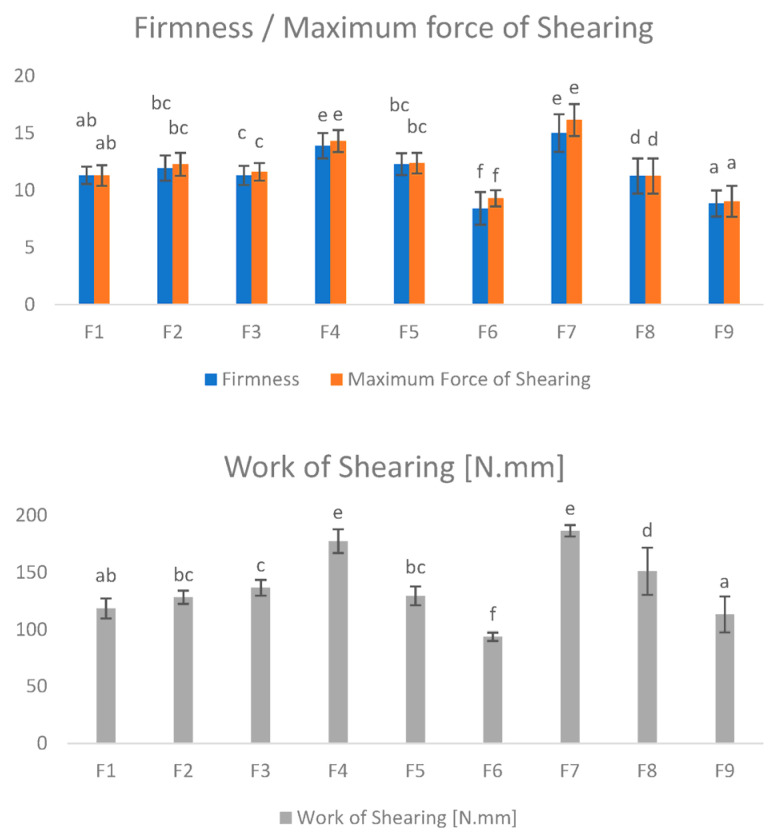
Warner–Bratzler analyses of the 9 hybrid meat samples. The samples, labelled from F1 to F9, contain different percentages of broccoli, upcycled brewer’s spent grain and insect flour (BR, BSG, IF): F1 (35, 0, 0), F2 (30, 5, 0), F3 (25, 0, 1), F4 (20, 5, 10), F5 (30, 0, 5), F6 (32, 3, 0), F7 (25, 5, 5), F8 (22, 3, 10) and F9 (27, 3, 5). Different letters in the same bar are significantly different according to Tukey’s test (*p* < 0.05).3.4. Modelling and Optimisation.

**Figure 5 foods-11-03396-f005:**
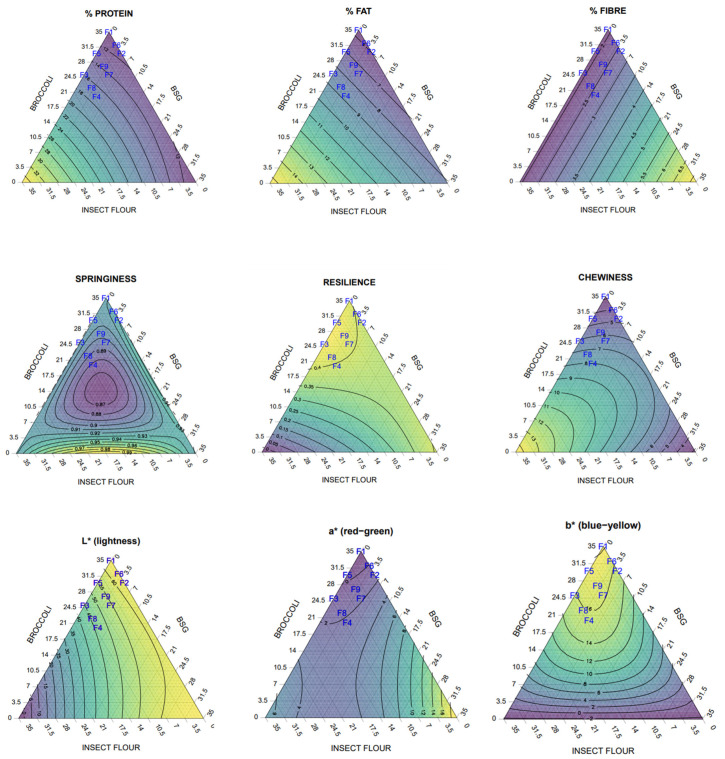
Ternary diagrams of the responses where R^2^ > 90%. The diagrams show the content of protein, fat and fibre content (percentage), the springiness, resilience and chewiness and the *L*a*b** colour parameters. The samples, labelled from F1 to F9, contain different percentages of broccoli, upcycled brewer’s spent grain and insect flour (BR, BSG, IF): F1 (35, 0, 0), F2 (30, 5, 0), F3 (25, 0, 1), F4 (20, 5, 10), F5 (30, 0, 5), F6 (32, 3, 0), F7 (25, 5, 5), F8 (22, 3, 10) and F9 (27, 3, 5).

**Figure 6 foods-11-03396-f006:**
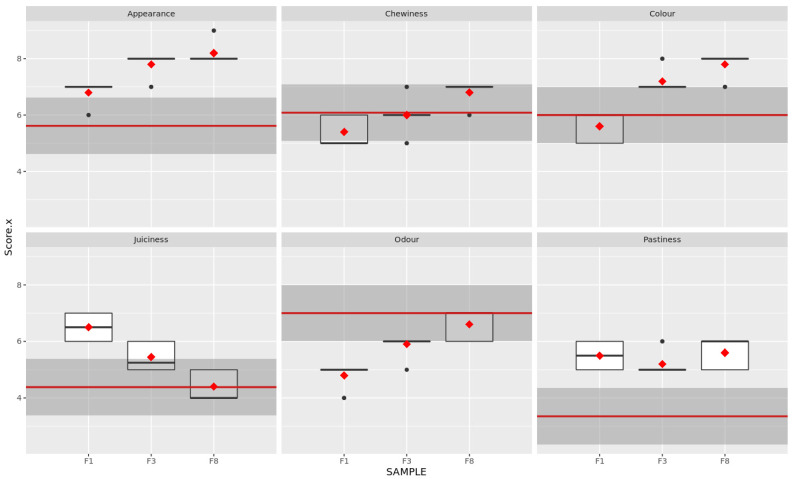
Boxplots for the attributes assessed for each sample (F1, F3 and F8) compared to the commercial sample QDA (horizontal red line). Dark grey area represents QDA values ± 1. Red dots represent the average of each panellist; black dots represent the outliers.

**Table 1 foods-11-03396-t001:** Physicochemical characteristics of upcycled brewer’s spent grain (BSG) and insect flour (IF). Values show mean ± standard deviation, n = 6.

Functional Properties	BSG	IF	*p*-Value
Moisture (%)	0.07 ^a^ ± 0.00	0.06 ^a^ ± 0.01	0.823
Swelling capacity (mL/g)	4.70 ± 0.10	3.40 ± 0.10	9.6 × 10^−05^ ***
Water solubility index (g/100 g)	0.52 ± 0.11	0.40 ± 0.05	0.188
Water holding capacity (g/g)	3.36 ± 0.08	3.02 ± 0.38	0.201
Oil holding capacity (g/g)	0.32 ± 0.01	0.37 ± 0.02	0.0212 *
a_w_	0.54 ± 0.01	0.61 ± 0.02	8.3 × 10^−16^ ***
Colour parameters			
*L**	57.22 ± 1.03	27.74 ± 8.04	1.5 × 10^−8^ ***
*a**	5.88 ± 0.22	8.92 ± 0.64	9.4 × 10^−10^ ***
*b**	24.38 ± 0.52	25.87 ± 0.92	0.000796 ***
Particle size characteristics			
d_10_ (µm)	103.93 ± 14.30	56.24 ± 23.89	9.9 × 10^−5^ ***
d_50_ (µm)	372.89 ± 19.46	481.56 ± 16.14	7.2 × 10^−10^ ***
d_90_ (µm)	766.44 ± 15.95	854.11 ± 6.23	5.4 × 10^−11^ ***
D[4.3] (µm)	407.67 ± 14.44	468.11 ± 16.29	3.3 × 10^−7^ ***
D[3.2] (µm)	140.4 ± 43.77	53.04 ± 12.35	2.9 × 10^−5^ ***
Specific Surface area	47.62 ± 17.98	118.43 ± 26.13	5.1 × 10^−6^ ***

Means with different superscript letters in the same column are significantly different according to Tukey’s test (*p* < 0.05). Significance codes for *p*-value: ***, (0 < *p*-value < 0.001); *, (0.01 < *p*-value < 0.05), (0.05 < *p*-value < 1).

**Table 2 foods-11-03396-t002:** Results for different formulations of hybrid sausages containing broccoli (BR), upcycled brewer’s spent grain (BSG) and insect flour (IF). Values show mean ± standard deviation, n = 27.

	BR (%)	BSG (%)	IF (%)	MC (%)	Protein (%)	Fat (%)	Fibre (%)	kcal/100 g	% kcal from Protein	Fe (%)	Zn (%)	*L**	*a**	*b**
F1	35	0	0	77.6 ^a^ ± 0.2	13.4 ^a^ ± 0.5	5.6 ^a^ ± 0.3	1.4 ^a^ ± 0.6	107.2 ^a^ ± 3.4	50.1 ^a^ ± 1.1	5.7 ^a^ ± 1.1	5.9 ^a^ ± 0.9	63.8 ^a^ ± 11.6	−1.6 ^e^ ± 0.3	17.9 ^a^ ± 3.9
F2	30	5	0	74.9 ^b^ ± 1.5	14.1 ^a^ ± 0.6	5.8 ^a^ ± 0.4	3.2 ^b^ ± 0.4	114.6 ^b^ ± 4.5	49.1 ^b^ ± 1.7	12.7 ^b^ ± 1.5	9.4 ^b^ ± 1.1	61.8 ^ab^ ± 1.9	1.4 ^bc^ ± 0.5	15.2 ^a^ ± 0.8
F3	25	0	10	70.1 ^c^ ± 1.3	17.6 ^b^ ± 0.5	8.4 ^b^ ± 0.2	1.8 ^a^ ± 0.5	149.8 ^a^ ± 3.5	47.0 ^a^ ± 0.8	11.7 ^b^ ± 1.0	21.0 ^c^ ± 1.4	56.7 ^ab^ ± 4.3	0.8 ^c^ ± 0.4	15.5 ^a^ ± 1.6
F4	20	5	10	66.5 ^d^ ± 1.9	18.1 ^b^ ± 0.7	8.7 ^b^ ± 0.0	4.5 ^b^ ± 0.3	159.9 ^a^ ± 7.6	45.3 ^a^ ± 1.5	16.2 ^c^ ± 1.1	23.0 ^c^ ± 1.2	54.3 ^b^ ± 2.9	2.2 ^a^ ± 0.3	15.6 ^a^ ± 0.8
F5	30	0	5	71.8 ^c^ ± 1.5	15.8 ^a^ ± 0.8	6.9 ^c^ ± 0.5	3.5 ^b^ ± 0.3	132.7 ^a^ ± 2.5	47.7 ^a^ ± 0.7	11.5 ^b^ ± 0.9	17.0 ^d^ ± 1.1	62.9 ^ab^ ± 4.2	−0.5 ^d^ ± 0.5	17.8 ^a^ ± 1.8
F6	32	3	0	72.3 ^c^ ± 1.4	16.2 ^c^ ± 1.0	6.0 ^c^ ± 0.5	3.6 ^b^ ± 0.3	126.5 ^a^ ± 8.7	51.4 ^a^ ± 1.5	11.9 ^b^ ± 1.2	15.0 ^d^ ± 0.9	64.5 ^a^ ± 3.8	−0.1 ^d^ ± 0.5	16.9 ^a^ ± 1.9
F7	25	5	5	70.6 ^c^ ± 1.1	16.2 ^c^ ± 0.7	7.5 ^a^ ± 0.2	3.9 ^b^ ± 0.2	139.8 ^b^ ± 5.8	46.2 ^b^ ± 0.8	14.4 ^d^ ± 0.8	18.0 ^d^ ± 0.9	58.8 ^ab^ ± 5.2	1.9 ^ab^ ± 0.7	16.0 ^a^ ± 1.5
F8	22	3	10	72.3 ^c^ ± 1.4	16.3 ^c^ ± 0.6	8.6 ^e^ ± 0.3	1.0 ^a^ ± 0.0	144.7 ^a^ ± 9.7	45.1 ^a^ ± 1.5	10.7 ^b^ ± 0.7	17.0 ^d^ ± 1.0	56.3 ^ab^ ± 3.9	1.7 ^ab^ ± 0.3	16.1 ^a^ ± 1.0
F9	27	3	5	72.7 ^c^ ± 1.7	16.1 ^c^ ± 0.7	7.3 ^d^ ± 0.0	1.8 ^a^ ± 0.0	133.5 ^a^ ± 8.9	48.2 ^a^ ± 1.1	10.8 ^b^ ± 0.8	15.0 ^d^ ± 1.1	62.1 ^ab^ ± 3.8	0.8 ^c^ ± 0.6	16.7 ^a^ ± 1.2

Means with different superscript letters in the same column are significantly different according to Tukey’s test (*p* < 0.05).

**Table 3 foods-11-03396-t003:** The texture profile (TPA) and Warner–Bratzler analyses (values show mean ± standard deviation, n = 27).

	Hardness (N)	Adhesiveness (N.mm)	Springiness	Cohesiveness	Chewiness (N)	Resilience	Work of Shearing (N.mm)	Shearing Force (N)
F1	3.51 ^a^ ± 0.39	0.15 ^ab^ ± 0.10	0.92 ^a^ ± 0.04	0.80 ^abc^ ± 0.01	2.56 ^a^ ± 0.21	0.40 ^abc^ ± 0.01	118.46 ^ab^ ± 8.76	11.32 ^a^ ± 0.76
F2	6.00 ^b^ ± 0.56	0.25 ^c^ ± 0.02	0.93 ^a^ ± 0.03	0.79 ^ab^ ± 0.04	4.43 ^b^ ± 0.67	0.39 ^a^ ± 0.02	128.30 ^bc^ ± 5.77	11.95 ^a^ ± 1.10
F3	7.93 ^c^ ± 1.31	0.13 ^ab^ ± 0.10	0.91 ^ab^ ± 0.02	0.81 ^abc^ ± 0.02	5.88 ^d^ ± 1.00	0.43 ^de^ ± 0.02	136.64 ^c^ ± 6.92	11.32 ^a^ ± 0.84
F4	10.97 ^d^ ± 0.80	0.09 ^b^ ± 0.03	0.87 ^b^ ± 0.02	0.79 ^a^ ± 0.02	7.61 ^ef^ ± 0.57	0.40 ^abc^ ± 0.02	177.51 ^e^ ± 10.51	13.92 ^c^ ± 1.11
F5	5.99 ^b^ ± 0.85	0.12 ^ab^ ± 0.06	0.92 ^a^ ± 0.03	0.82 ^c^ ± 0.02	4.59 ^bc^ ± 0.62	0.45 ^e^ ± 0.02	129.49 ^bc^ ± 8.21	12.30 ^a^ ± 0.96
F6	6.07 ^b^ ± 2.60	0.13 ^ab^ ± 0.08	0.93 ^a^ ± 0.04	0.79 ^ab^ ± 0.02	4.49 ^b^ ± 2.03	0.40 ^ab^ ± 0.03	93.54 ^f^ ± 3.68	8.42 ^b^ ± 1.44
F7	8.96 ^c^ ± 0.99	0.20 ^bc^ ± 0.11	0.90 ^ab^ ± 0.05	0.79 ^ab^ ± 0.01	6.44 ^de^ ± 0.77	0.41 ^abc^ ± 0.01	186.55 ^e^ ± 5.00	15.03 ^c^ ± 1.64
F8	11.03 ^d^ ± 0.55	0.12 ^ab^ ± 0.03	0.90 ^ab^ ± 0.04	0.80 ^abc^ ± 0.01	7.95 ^f^ ± 0.58	0.41 ^bcd^ ± 0.01	151.10 ^d^ ± 20.77	11.26 ^a^ ± 1.54
F9	7.86 ^c^ ± 1.38	0.19 ^abc^ ± 0.09	0.90 ^ab^ ± 0.04	0.81 ^bc^ ± 0.01	5.73 ^cd^ ± 1.22	0.42 ^cd^ ± 0.01	113.28 ^a^ ± 15.76	8.86 ^b^ ± 1.14

Means with different superscript letters in the same column are significantly different according to Tukey’s test (*p* < 0.05).

**Table 4 foods-11-03396-t004:** Summary of statistics for the multiple linear regression, showing R-squared fit, F-value, and *p*-value for the model and for each independent variable and their interactions.

	Protein	Fat	Fibre	Fe	Zn	Hardness (N)	Adhesiveness (N.mm)	Springiness	Cohesiveness	Chewiness (N)	Resilience	Work of Shearing (N.mm)	Shearing Force (N)	*L**	*a**	*b**
R^2^	0.996	0.991	1.000	0.737	0.723	0.757	0.765	0.935	0.844	0.960	0.952	0.825	0.853	0.933	0.991	0.912
F	388.954	321.551	38656.325	4.682	7.837	5.179	5.429	23.794	9.050	40.445	33.068	7.841	9.688	23.199	187.545	17.356
Pr > F	<0.0001	<0.0001	<0.0001	0.065	0.021	0.054	0.050	0.002	0.018	0.001	0.001	0.025	0.016	0.002	<0.0001	0.004
BR		632.763	23.688		15.014										246.616	42.049
		<0.0001	0.003		0.008										<0.0001	0.001
BSG		71.183	60584.748	7.678	1.033	12.228	8.437					7.221	20.878			
		0.0002	<0.0001	0.039	0.349	0.017	0.034					0.043	0.006			
IF	51.711								7.668	34.040	26.307			11.284	134.344	
	0.001								0.039	0.002	0.004			0.020	<0.0001	
BR*BSG	26.358					12.297		4.023	4.884	11.952	9.704	7.223	20.843	1.427		
	0.004					0.017		0.101	0.078	0.018	0.026	0.043	0.006	0.286		
BR*IF	1.657			4.134					8.236		32.778			3.969		22.921
	0.254			0.098					0.035		0.002			0.103		0.005
BSG*IF							4.474	0.080				3.976	16.302		25.766	
							0.088	0.788				0.103	0.010		0.004	
BR*BSG*IF				0.971		8.760	2.709	1.878		0.678						20.657
				0.370		0.032	0.161	0.229		0.448						0.006

**Table 5 foods-11-03396-t005:** Results of the tasting sessions (n = 3) where samples REF, F1, F3 and F8 were assessed. F-value and *p*-value show the statistics of significance. Significance codes for *p*-value: ***, (0 < *p*-value < 0.001). Means with different superscript letters in the same column are significantly different according to Tukey’s test (*p* < 0.05).

Samples	BR (%)	BSG (%)	IF (%)	Appearance	Chewiness	Colour	Juiciness	Odour	Pastiness
F8	22	3	10	8.13 ^a^	6.87 ^a^	7.80 ^a^	4.43 ^c^	6.60 ^a^	5.67 ^a^
REF	-	-	-	5.62 ^c^	6.08 ^b^	6.00 ^c^	4.38 ^c^	7.00 ^a^	3.35 ^b^
F3	25	0	10	7.73 ^a^	6.07 ^b^	7.20 ^b^	5.48 ^b^	5.90 ^b^	5.27 ^a^
F1	35	0	0	6.73 ^b^	5.47 ^c^	5.60 ^c^	6.58 ^a^	4.80 ^c^	5.57 ^a^
F value				89.24	15.13	81.46	45.2	39.72	65.88
*p*-value				<2 × 10^−16^ ***	8.9 × 10^−07^ ***	<2 × 10^−16^ ***	4.6 × 10^−13^ ***	3.3 × 10^−12^ ***	9.9 × 10^−16^ ***

## Data Availability

The data used to support the findings of this study can be made available by the corresponding author upon request.

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
