# Peer review of "Hybrid Sausages: Modelling the Effect of Partial Meat Replacement with Broccoli, Upcycled Brewer’s Spent Grain and Insect Flours"

_foods, 2022, doi:10.3390/foods11213396_

Round 1

Reviewer 1 Report

The topic that the article presents is very current and practically usable. Hybrid meat products can be a solution to overcome the reluctance of part of the population to give up meat products while reducing meat production and consumption. The special value of the presented research lies in finding a suitable combination of three alternative raw materials, which will provide the hybrid product with properties fully comparable to the traditional meat product. Methods for the raw materials combination selection are adequate to obtain the desired outputs and are sufficiently described. The article lacks a detailed chemical analysis of products nutrients (profile of amino acids, fatty acids), which the authors themselves mention in the conclusion part and indicate as further research directions.

Minor modifications:

Fig. 3: percentage of the replacements in all samples is 35% except F3: Br, GSG, IF = (25,0,1). According to the design plot should be (25, 0, 10)

line 160: "9 assays" instead of "9 essays"

Author Response

The topic that the article presents is very current and practically usable. Hybrid meat products can be a solution to overcome the reluctance of part of the population to give up meat products while reducing meat production and consumption. The special value of the presented research lies in finding a suitable combination of three alternative raw materials, which will provide the hybrid product with properties fully comparable to the traditional meat product. Methods for the raw materials combination selection are adequate to obtain the desired outputs and are sufficiently described. The article lacks a detailed chemical analysis of products nutrients (profile of amino acids, fatty acids), which the authors themselves mention in the conclusion part and indicate as further research directions. Minor modifications: Fig. 3: percentage of the replacements in all samples is 35% except F3: Br, GSG, IF = (25,0,1). According to the design plot should be (25, 0, 10) line 160: "9 assays" instead of "9 essays" The authors highly appreciate the valuable comments from the reviewer. The suggested minor changes have been now addressed in the manuscript.

Reviewer 2 Report

1.       Please italicize L* a* and b* throughout the text.

2.       How about the storage stability of the hybrid sausage when compare to the conventional Bratwurst? Due to the different compositions, the oxidative stability and microbial quality should be concerned.

3.       Did the authors adjust protein to lipid ratio of each formulae? If not why?

4.       In the Introduction. State more reason why broccoli was used instead of other vegetables like cereals, legumes and nuts.

5.       Can you suggest the pre-treatment of the broccoli to avoid the texture and color defect?

6.       For all bar charts, please add the letters to indicate significant different on the bar.

Author Response

The authors highly appreciate the valuable comments from the reviewer, that have increased the quality of the manuscript. Please find below point-by-point response:

  1. Please italicize L* a* and b* throughout the text.

Format has been changed.

  1. How about the storage stability of the hybrid sausage when compare to the conventional Bratwurst? Due to the different compositions, the oxidative stability and microbial quality should be concerned.

The sausages were pasteurised following the standard procedure for cooked sausages. Although the data is not shown in the text. A temperature probe was used to ensure that the core of the sausage reached at least 85ºC during 17 min for all batches. The F-value was 1200 min. After, the sausages were placed in a blast chiller, until they reached 4ºC. They were then stored in the freezer at -18ºC. Data registered by the temperature probe is provided for revision purposes only.

  1. Did the authors adjust protein to lipid ratio of each formulae? If not why?

The protein:lipid ratio was not adjusted because the fat content of the ingredients (specially insect flour) would affect the samples where this ingredient was present.

However, the authors are preparing another manuscript showing these results. It was interesting to see how insect fat profile affected the fatty acid profile of the hybrid sausage in different ways.

  1. In the Introduction. State more reason why broccoli was used instead of other vegetables like cereals, legumes and nuts.

The following reason has been added:

Broccoli (Brassica oleracea var. italica) Broccoli offers a suitable combination of vita-mins and minerals, and it can compete with beef and chicken eggs in protein composition and tryptophan and lysine content [18]. Its bioactive phytochemical and nutritional richness, including phenolic and organosulphur compounds, as well as essential vitamins and minerals has called the attention of the scientific community. Furthermore, the broccoli fibre and fibre fetlock regulate the functioning of the gastrointestinal tract.

  1. Can you suggest the pre-treatment of the broccoli to avoid the texture and color defect?

Broccoli was bought from the supermarket in a frozen state to standardize the procedure for sample preparation, and avoid differences among batches. This has now been stated in the text.

  1. For all bar charts, please add the letters to indicate significant different on the bar.

Significant letters have been added in all bar charts.
